# Relationships between Maternal Gene Polymorphisms in One Carbon Metabolism and Adverse Pregnancy Outcomes: A Prospective Mother and Child Cohort Study in China

**DOI:** 10.3390/nu14102108

**Published:** 2022-05-18

**Authors:** Shuxia Wang, Yifan Duan, Shan Jiang, Ye Bi, Xuehong Pang, Changqing Liu, Zhenyu Yang, Jianqiang Lai

**Affiliations:** 1Key Laboratory of Trace Element Nutrition of National Health Commission, National Institute for Nutrition, Health Chinese Center for Disease Control and Prevention, Beijing 102206, China; wangsx@ninh.chinacdc.cn (S.W.); duanyf@ninh.chinacdc.cn (Y.D.); jiangshan@ninh.chinacdc.cn (S.J.); biye@ninh.chinacdc.cn (Y.B.); pangxh@ninh.chinacdc.cn (X.P.); laijq@ninh.chinacdc.cn (J.L.); 2Hebei Provincial Center for Disease Control and Prevention, Shijiazhuang 050021, China; lcq93@126.com

**Keywords:** polymorphism, *MTHFR*, *MTRR*, *MTR*, *TYMS*, preterm delivery, low birth weight, small-for-gestational-age

## Abstract

Background: To investigate relationships between five single nucleotide polymorphisms (SNP) in four maternal genes involved in one carbon metabolism and adverse pregnancy outcomes, including preterm birth (PTB), low birth weight (LBW), and small-for-gestational-age (SGA). Methods: This was a prospective mother and child cohort study in Wuqiang, China. Pregnant women (*n* = 939) were recruited from Jun 2016 to Oct 2018. Pregnancy outcomes (PTB, LBW, and SGA) were extracted from medical records and other information including age at childbearing, maternal education level, gravidity, parity, pre-pregnancy weight and height was collected by using a structured questionnaire. The maternal serum folate concentration was measured by using Abbott Architect i2000SR chemiluminescence analyzer in the first prenatal care visit. DNA genotyping of methylenetetrahydrofolate reductase (*MTHFR*) C677T and A1298C, methionine synthase reductase (*MTRR*) A66G, methionine synthase (*MTR*) A2756G, and thymidylate synthetase (*TYMS*) rs3819102 was processed by Sequenom MassARRAY iPLEX Platform. Univariate and multivariate logistics regression analysis were used to test the relationships between 5 SNPs and PTB, LBW, SGA. Results: Totally, 849 dyads of women and infants were included in the analysis. The prevalence of PTD, LBW, and SGA were 3.76%, 1.58%, and 5.31% respectively. The homozygote frequencies of *MTHFR* C677T, *MTHFR* A1298C, *MTRR* A66G, *MTR* A2756G, and *TYMS* rs3819102 were 44.2%, 1.4%, 6.7%, 1.3%, and 3.2%, and the alt allele frequencies were 66.1%, 10.8%, 24.9%, 10.5%, and 20.5% respectively. The average serum folate concentration was 11.95 ng/mL and the folate deficiency rate was 0.47%. There were no significant associations between *MTHFR* C677T, *MTHFR* A1298C, *MTRR* A66G, *MTR* A2756G, *TYMS* rs3819102 alleles and PTD, LBW, SGA (*p* > 0.05). Conclusions: In the population with adequate folate status and low prevalence of adverse pregnancy outcomes, *MTHFR* C677T, *MTHFR* A1298C, *MTRR* A66G, *MTR* A2756G, TYMS rs3819102 alleles may not be related to PTD, LBW, and SGA.

## 1. Introduction

Adverse pregnancy outcomes, including preterm birth (PTB), low birth weight (LBW), and small-for-gestational-age (SGA) are major determinants for infant morbidity and mortality. PTB and its complications are the leading causes of death among children under 5 years old [1]. World Health Organization (WHO) estimated that 14.84 million babies were born preterm globally in 2014 and there were the second highest numbers of estimated preterm births in China [2]. LBW was associated with physical and mental development retardation and higher rate of children mortality. According to the estimates from UNICEF and WHO, 20.5 million LBW babies were born globally in 2015, and nearly 8.47 million LBW babies were born in China [3]. SGA posted a huge burden worldwide. In 2010, an estimated 32.4 million infants were born SGA in low and middle income countries [4]. China Nutrition and Health Surveillance on 0–5 years Children and Lactating Women in 2013 reported that the prevalence of PTB and LBW were 9.9% and 3.0%~4.0% in 0~ to 5~ years old, respectively in 2013 [5]. The burden of undernutrition including PTB, LBW, and SGA is heavy globally, especially in developing countries [6,7].

Environmental factors were related to PTB, LBW, and SGA, including behavioral and sociodemographic factors (e.g., maternal education level, gestational age, alcohol, and smoking consumption) and medical and pregnancy conditions (e.g., gestational hypertension, gestational diabetes mellitus) [8,9,10], and genetic factors could be associated to PTB, LBW, and SGA too. Some evidence suggested that genetic factors played an important role in the occurrence of PTB, LBW, and SGA [11,12,13]. As a one-carbon metabolism enzyme, folic acid is critical in DNA and RNA synthesis. High-level folate was associated with lower risks of the above adverse pregnancy outcomes [14,15]. Single nucleotide polymorphisms (SNPs)-related folic acid metabolic enzymes in birth defects had been identified [16]. Recently, both Indian and Japanese studies suggested that folic acid metabolic enzymes-related SNPs may be related to the risks of PTB and LBW [17,18]. However, the existing SNP studies in PTB, LBW, and SGA were limited and the results were inconsistent [19,20,21,22,23,24,25,26,27]. A study showed that the lower level of folate and unfavorable mutations contributed to PTB, and some other outcomes including hyperhomocysteinaemia [28]. Methylenetetrahydrofolate reductase (*MTHFR*) C677T and A1298C, methionine synthase reductase (*MTRR*) A66G, methionine synthase (*MTR*) A2756G, thymidylate synthetase (*TYMS*) rs3819102 were the common target SNPs in the previous studies. 

Therefore, we aimed to investigate relationships between the five SNPs in four genes (*MTHFR*, *MTRR*, *MTR,* and *TYMS*) involved in one carbon metabolism, folic acid status, and adverse pregnancy outcomes (PTB, LBW, and SGA) in a Chinese mother and child cohort.

## 2. Materials and Methods

### 2.1. Study Subjects

A prospective maternal and child nutrition and health cohort in China was carried out in Wuqiang, China [29]. All pregnant women were recruited in the cohort during Jun 2016 to Oct 2018 at prenatal care center and were followed through delivery in the current study. Inclusion criteria: (1) pregnant women aged 18 to 45 years old, (2) gestational week < 20, (3) singleton pregnancy. Exclusion criteria: pregnant women with history of habitual abortion, diabetes, hypertension, and thrombophilia. Totally, 939 pregnant women were enrolled in the study. The study was approved by the National Institute for Nutrition and Health, Chinese Center for Disease Control and Prevention Ethical Review Committee, and all study subjects provided written informed consent.

### 2.2. Data Collection

A structured questionnaire was used to collect maternal sociodemographic information at enrollment, including age at childbearing, maternal education level, gravidity, parity, pre-pregnancy weight and height. Maternal health and pregnancy outcomes including PTB, LBW, and SGA were extracted from pregnancy examination records and delivery medical records. PTB defined childbirth before 37 weeks of pregnancy, and LBW defined birth weight less than 2500 g [30]. SGA defined birthweight less than the tenth percentile at a particular gestational week [31]. The gestational week and birth weight data of baby were extracted from delivery medical records. For those women delivering at non-study hospital, women were asked to recall the delivery information, including gestational age at delivery, gender and birth weight of infant.

### 2.3. Laboratory Analysis

Venous blood samples were collected in the first prenatal care visit, and serum was separated by centrifugation and stored in −20 ℃ freezer. Serum folate concentration was measured by using Abbott Architect i2000SR chemiluminescence analyzer with folate testing kit (Abbott, Shanghai, China, 40320–40321). Liquichek Immunoassay Plus Control (Bio-Rad Laboratories, Shanghai, China, 40320) was used as quality control. Mean intra-assay and inter-assay coefficient of variations (CV) for serum folate concentration measurements ranged from 4.48% to 7.29% and 6.13% to 11.37% for low level, 2.79% to 3.14% and 4.18 to 4.19% for medium level, 2.59% to 3.16% and 4.23% to 4.24% for high-level, respectively. Serum folate concentration < 2 ng/mL was defined as folate deficiency.

DNA was extracted from 1 milliliter whole blood by using TIANGEN TIANamp Blood DNA Kit (TIANGEN BIOTECH CO., LTD., Beijing, DP349-02). The *MTHFR*, *MTRR*, *MTR,* and *TYMS* polymorphisms were genotyped by using Sequenom MassARRAY iPLEX Platform (Agena Bioscience, San Diego, CA, USA) [32]. DNA concentration ≥10 ng/ul and volume ≥20 ul are required for the assay. The assay consisted of an initial PCR reaction, followed by single base extension using mass-modifies dideoxynucleotide terminators of an oligonucleotide primer. SNPs were genotyped by using MALDI-TOF mass spectrometry and classified as “A. Conservative”, “B. Moderate”, “C. Aggressive” by the spectrometry automatically or “D” artificially by the lab technician if the spectrometry cannot assign the one into a category defined previously. 

### 2.4. Statistical Analysis

All data were exported from the data collection system and imported into SAS 9.4 for statistical description and analysis. The missing values of maternal age at childbearing, education level, pre-pregnancy weight and height were imputed by PROC MI. Data were described as the mean and standard deviation (SD) or median for continuous variables and frequencies (%) for categorical variables. Univariate and multivariate logistics regression analysis were used to test the associations between PTB, LBW, SGA and *MTHFR*, *MTRR*, *MTR*, *TYMS* polymorphisms. In the multivariate logistics model, maternal education, maternal age at childbearing, pre-pregnancy BMI, serum folate concentration, and gender of baby were adjusted as covariates only if they could alter 10% of the OR value in 5 SNPs respectively. 

Sensitive analyses were carried out in women whose SNPs genotyped description of “A. Conservative” or whose delivery information was available only from delivery medical records respectively. In the subgroup of “A. Conservative genotype”, the cases included women with adverse pregnancy outcomes (PTB/LBW/SGA) and the controls included those without those adverse pregnancy outcomes. In the subgroup of “delivery medical records”, the cases included women with adverse pregnancy outcomes (PTB/LBW/SGA) based on medical records and the controls included those without adverse pregnancy outcomes based on medical records.

## 3. Results

### 3.1. Demographic Characteristics

Of the 939 pregnant women, 18 women had abortion, induced labor, or dead fetus. After excluding those subjects with miscarriage, still birth, or twin pregnancy, there remained 911 women in the study. Of the 911 women, 849 women had genotyping results and serum folate concentration, and 62 women were excluded because DNA concentration was less than 10 ng/ul (Figure 1). 

The average maternal age at childbearing in the study was 28.4 years old and gestational age at delivery was 39.0 weeks. About 8.25% pregnant women had college degree. The average pre-pregnancy BMI was 23.67 kg/m^2^. The average serum folate concentration was 11.95 ng/mL and the folate deficiency rate was 0.47%. Of the 849 pregnant women, 718, 697, and 697 had PTB, LBW, and SGA results respectively. The proportion of results coming from medical records were 74.5% (535/718), 76.8% (535/697), and 76.8 (535/697) respectively. The prevalence of PTD, LBW, and SGA were 3.76%, 1.58%, and 5.31% respectively. The characteristics of study population divided by PTB, LBW, and SGA are shown in Table 1. 

### 3.2. Genotype Frequencies of SNP

The homozygote frequencies of *MTHFR* C677T, *MTHFR* A1298C, *MTRR* A66G, *MTR* A2756G, and *TYMS* rs3819102 are 44.2%, 1.4%, 6.7%, 1.3%, and 3.2%, and the alt allele frequencies are 66.1%, 10.8%, 24.9%, 10.5%, and 20.5% respectively. The genotype frequencies of *MTHFR* C677T, *MTHFR* A1298C, *MTRR* A66G, *MTR* A2756G, and *TYMS* rs3819102 have no deviation from the Hardy–Weinberg equilibrium (HWE) in all the pregnant women (Table 2).

### 3.3. Associations between SNPs and Adverse Pregnancy Outcomes

No significant associations were found between *MTHFR* C677T (PTB: OR 1.14, 95% CI 0.34–3.86; SGA: OR 0.71, 95% CI 0.29–1.76), *MTHFR* A1298C (PTB: OR 0.86, 95% CI 0.32–2.30; LBW: OR 0.37, 95% CI 0.05–2.91; SGA: OR 0.88, 95% CI 0.38–2.03), *MTRR* A66G (PTB: OR 0.68, 95% CI 0.30–1.54; LBW: OR 0.79, 95% CI 0.23–2.71; SGA: OR 0.66, 95% CI 0.32–1.33), *MTR* A2756G (PTB: OR 1.82, 95% CI 0.78–4.24; LBW: OR 1.57, 95% CI 0.41–6.01; SGA: OR 1.17, 95% CI 0.52–2.62), *TYMS* rs3819102 alleles (PTB: OR 0.84, 95% CI 0.37–1.91; LBW: OR 0.37, 95% CI 0.08–1.72; SGA: OR 0.53, 95% CI 0.25–1.14) and either of PTD, LBW, and SGA (*p* > 0.05). (Table 3, Table 4 and Table 5)

After adjusting for maternal education, maternal age at childbearing, pre-pregnancy BMI, serum folate concentration and gender of baby, the relationships remained non-significant between *MTHFR* C677T, *MTHFR* A1298C, *MTRR* A66G, *MTR* A2756G, *TYMS* rs3819102 alleles and PTD, LBW, SGA (*p* > 0.05).

The results of sensitive analyses were in accordance with the full analysis (Appendix A).

## 4. Discussion

We found that five maternal SNPs in *MTHFR*, *MTRR*, *MTR*, and *TYMS* were not associated with adverse pregnancy outcomes (PTD, LBW, and SGA) in the current study.

The alt allele frequency of *MTHFR* C677T, *MTHFR* A1298C, *MTRR* A66G, *MTR* A2756G, *TYMS* rs3819102 was 66.1%, 10.8%, 24.9%, 10.5%, and 20.5% in the current study, respectively. The frequency of SNP varied greatly among different races. According to the Allele Frequency Aggregator, the alt allele frequency of *MTHFR* C677T, *MTHFR* A1298C, *MTRR* A66G, *MTR* A2756G, *TYMS* rs3819102 was 34.0%, 30.4%, 51.7%, 19.1%, 2.6% globally and 38.6%, 21.4%, 27.3%, 10.8%, 23.6% in the East Asian respectively [33,34,35,36,37]. Compared to the East Asian population, the alt allele frequency of *MTHFR* C677T was higher and *MTHFR* A1298C lower in the current study. The frequency of other SNPs was similar as East Asian level. 

Some studies suggested that the level of maternal serum folate was negatively related to PTB, LBW, and SGA [14,15,38]. In our study, the average serum folate concentration was 11.95 ng/mL and the folate deficiency rate was 0.47%, which may protect from these adverse pregnancy outcomes in the current study. 

Two studies showed no significant association between *MTHFR* C677T, *MTHFR* A1298C and PTB, SGA in either whites or blacks in the presence of high folate intake after mandatory grain folic fortification in the United State [39,40], even in low folate intake subgroup status, some other factors seemed to affect the association between *MTHFR* SNPs and adverse pregnancy outcomes, such as racial difference [39]. Nurk’s study found no significant associations between *MTHFR* C677T and LBW in Norway [41]. Moreover, the Screening for Pregnancy Endpoints (SCOPE) prospective cohort study found no significant gene-nutrient interactions between maternal *MTHFR* C677T, *MTHFR* A1298C, and folic acid use, and their association with PTB and SGA [42,43]. The results of the three studies mentioned above were collaborative with our study. However, a case-control study nested in a multicenter cohort found *MTHFR* C677T was positively associated with PTB and LBW in India, in which folate status was not reported. Extremely, very, and moderately PTB showed higher frequency of *MTHFR* C677T mutation compared to term delivery cases [20]. A Japanese study suggested that maternal *MTHFR* C677T, not *MTHFR* A1298C was independently associated with improvement in infant birth weight in the background of 16.4 nmol/L serum folate [25]. The inconsistence may be related to differences in dietary folate status or mean birth weight. Although hyperhomocysteinemia induced by *MTHFR* variant genotypes have shown a strong relationship with adverse pregnancy outcomes, high-level intake of folate could reverse the results [44]. In our study, folic acid status was adequate. The results suggested sufficient folic acid status could be crucial to the pregnancy outcomes, including PTB, LBW, and SGA, especially for those with SNPs mutation. 

*MTRR* and *MTR* are the key enzymes in the one-carbon pool by folate, catalyzing the methylation of homocysteine to methionine. The SCOPE study found no significant gene-nutrient interactions between maternal *MTRR* A66G, *MTR* A2756G, and folic acid use, and their association with PTB and SGA [42,43], which aligned with our finding of *MTRR* and *MTR* SNPs. Tiwari found that *MTRR* A66G was not related to PTB on the background of high-level folate, but homocysteine concentration did [20]. Another nested case-control study carried out by Engel found that *MTRR* A66G was positively related to PTB in the white population, but not in the black population regardless of folate intake level [39]. *MTRR* A66G and *MTR* A2756G have been proved to be associated with increased homocysteine levels [45,46], and a certain concentration of homocysteine could cause endothelial cell dysfunction and oxidative stress, which is positively related to PTB, LBW, and preeclampsia [47]. Meanwhile, no association was found between *MTRR* A66G and SGA [39]. However, we found no association between either *MTRR* or *MTR* variants associated with PTB, LBW, and SGA in our study. 

*TYMS* catalyzes the methylation of deoxyuridylate to deoxythymidylate, which maintains the dTMP pool critical for DNA replication and repair [48]. There were few studies involving the associations between *TYMS* rs3819102 gene polymorphism and PTB, LBW, SGA. We first explored the association between *TYMS* rs3819102 and adverse pregnancy outcomes.

The current study investigated the relationships between five SNPs in four maternal genes involved in one carbon metabolism and adverse pregnancy outcomes, including PTB, LBW, and SGA, and folate status was taken into consideration for the relationship simultaneously. The limitation of this study was the high loss to follow-up rate. Many pregnant women did not follow-up because of delivering at a different hospital, and could not be reached by telephone. However, the results of sensitive analyses were in accordance with the full analysis, which suggested loss to follow-up rate did not affect the results. Moreover, this study did not analyze epigenetics of study population, which could modify the gene expression [49] and the interaction between gene polymorphism and epigenetics may alter the activity of folic acid-related enzymes. However, adequate folic acid status of those pregnant women may maintain similar methylation level of those subjects, which may reduce the impacts of epigenetics.

## 5. Conclusions

In the population with adequate folic acid status and low prevalence of adverse pregnancy outcomes, MTHFR C677T, MTHFR A1298C, MTRR A66G, MTR A2756G, TYMS rs3819102 alleles may not be related to PTD, LBW, and SGA. It is critical to maintain good folate status for preventing adverse pregnancy outcomes including PTB, LBW, and SGA, even for those with one carbon metabolism enzyme mutations.

## Figures and Tables

**Figure 1 nutrients-14-02108-f001:**
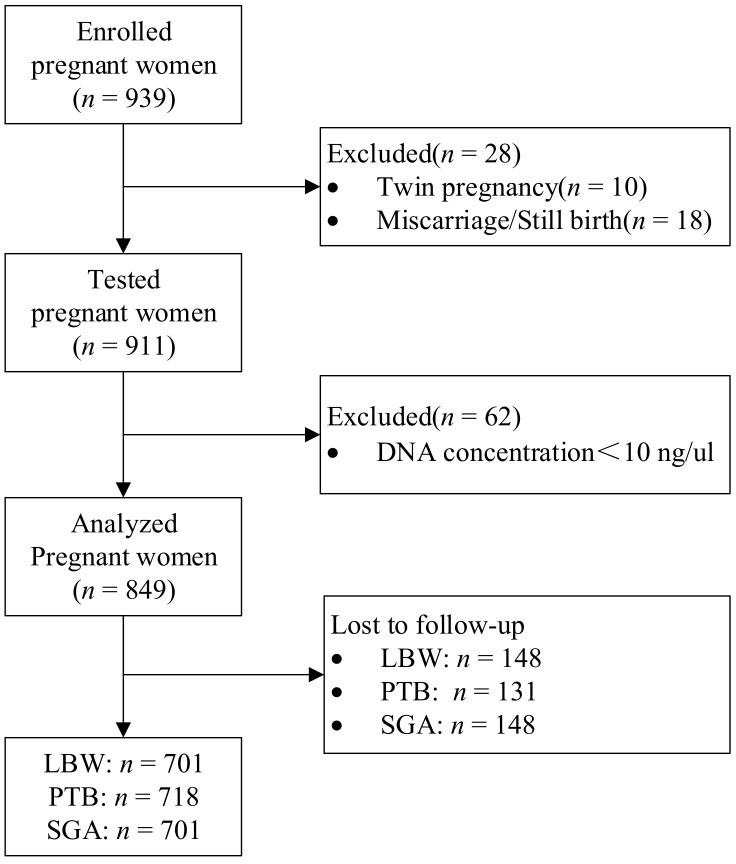
Flow chart of the study population.

**Table 1 nutrients-14-02108-t001:** Characteristics of study population.

	PTB		LBW		SGA	
	Cases (*n* = 27)	Controls (*n* = 691)	Cases (*n* = 11)	Controls (*n* = 686)	Cases (*n* = 37)	Controls (*n* = 660)
Age at childbearing, mean (SD), y	28.5(2.5)	28.4(4.0)	28.2(3.0)	28.4(3.9)	28.5(5.0)	28.4(3.9)
Maternal education, *n* (%)						
≥College degree	5(18.52)	56(8.24)	2(18.18)	59(8.60)	0(0)	60(9.09)
Middle school degree	19(70.37)	599(88.08)	7(63.64)	598(88.05)	9(94.59)	569(87.27)
<Middle school degree	3(11.11)	25(3.68)	2(18.18)	23(3.35)	2(5.41)	24(3.64)
Pre-pregnancy BMI, mean (SD), kg/m^2^	24.26(5.07)	23.72(4.39)	25.43(4.45)	23.70(4.40)	21.64(5.11)	23.72(4.31)
Gravidity, *n* (%)						
1	9(33.33)	174(25.22)	5(45.45)	171(24.96)	13(35.14)	163(24.73)
≥2	18(66.67)	516(74.78)	6(54.55)	514(75.04)	24(64.86)	496(75.27)
Serum folate concentration, mean (SD), ng/mL	11.96(3.25)	11.98(5.10)	11.17(2.74)	12.01(5.11)	11.26(5.17)	12.08(5.06)
Folate deficiency, *n* (%)	0(0)	4(0.57)	0(0)	4(0.58)	0(0)	4(0.60)
Gestational age at delivery, mean (SD), wk	34.7(1.8)	39.1(0.9)	35.3(2.5)	39.0(1.1)	39.0(1.3)	39.0(1.2)
Gender of infant, *n* (%)						
Boy	16(59.26)	334(48.34)	4(36.36)	331(48.25)	16(43.24)	320(48.48)
Girl	11(40.74)	357(51.66)	7(63.64)	355(51.75)	21(56.76)	340(51.52)
Birth weight, mean (SD), g	2656(502)	3406(406)	2104(235)	3399(403)	2557(222)	3406(408)

PTB, preterm birth; LBW, low birth weight; SGA, small-for-gestational-age.

**Table 2 nutrients-14-02108-t002:** The frequency of genotypes and alleles of *MTHFR*, *MTRR*, *MTR*, *TYMS*.

Gene	Genotype [*n*(%)]			Allele [*n*(%)]		P(HWE)
	Wild Type	Heterozygote	Homozygote	Ref Allele	Alt Allele	
*MTHFR* C677T	102(12.0)	372(43.8)	375(44.2)	576(33.9)	1122(66.1)	0.540
*MTHFR* A1298C	678(79.9)	159(18.7)	12(1.4)	1515(89.2)	183(10.8)	0.473
*MTRR* A66G	483(57.0)	308(36.3)	57(6.7)	1274(75.1)	422(24.9)	0.408
*MTR* A2756G	682(80.3)	156(18.4)	11(1.3)	1520(89.5)	178(10.5)	0.577
*TYMS* rs3819102	528(62.2)	294(34.6)	27(3.2)	1350(79.5)	348(20.5)	0.073

*MTHFR*, methylenetetrahydrofolate reductase; *MTRR*, methionine synthase reductase; *MTR*, methionine synthase; *TYMS*, thymidylate synthetase; HWE, Hardy–Weinberg equilibrium.

**Table 3 nutrients-14-02108-t003:** Relationship between genotypes of MTHFR, MTRR, MTR, TYMS, and PTB.

Genotype		Controls [*n*(%)]	PTB [*n*(%)]	OR (95% CI)	*p*
*MTHFR* C677T	CC	86(96.63)	3(3.37)	1	-
	CT + TT	605(96.18)	24(3.82)	1.14(0.34–3.86)	0.837
*MTHFR* A1298C	AA	546(96.13)	22(3.87)	1	-
	AC + CC	145(96.67)	5(3.33)	0.86(0.32–2.30)	0.758
*MTRR* A66G	AA	399(95.68)	18(4.32)	1	-
	AG + GG	292(97.01)	9(2.99)	0.68(0.30–1.54)	0.359
*MTR* A2756G	AA	561(96.72)	19(3.28)	1	-
	AG + GG	130(94.20)	8(5.80)	1.82(0.78–4.24)	0.167
*TYMS* rs3819102	AA	434(96.02)	18(3.98)	1	-
	AG + GG	257(96.62)	9(3.38)	0.84(0.37–1.91)	0.684

*MTHFR*, methylenetetrahydrofolate reductase; *MTRR*, methionine synthase reductase; *MTR*, methionine synthase; *TYMS*, thymidylate synthetase; PTB, preterm birth.

**Table 4 nutrients-14-02108-t004:** Relationship between genotypes of *MTHFR*, *MTRR*, *MTR*, *TYMS,* and LBW.

Genotype		Controls [*n*(%)]	LBW [*n*(%)]	OR (95% CI)	*p*
*MTHFR* C677T	CC	84(100.00)	0(0)	1	-
	CT + TT	602(98.21)	11(1.79)	-	0.973
*MTHFR* A1298C	AA	540(98.18)	10(1.82)	1	-
	AC + CC	146(99.32)	1(0.68)	0.37(0.05–2.91)	0.345
*MTRR* A66G	AA	397(98.27)	7(1.73)	1	-
	AG + GG	289(98.63)	4(1.37)	0.79(0.23–2.71)	0.702
*MTR* A2756G	AA	554(98.58)	8(1.42)	1	-
	AG + GG	132(97.78)	3(2.22)	1.57(0.41–6.01)	0.507
*TYMS* rs3819102	AA	428(97.94)	9(2.06)	1	-
	AG + GG	258(99.23)	2(0.77)	0.37(0.08–1.72)	0.204

*MTHFR*, methylenetetrahydrofolate reductase; *MTRR*, methionine synthase reductase; *MTR*, methionine synthase; *TYMS*, thymidylate synthetase; LBW, low birth weight.

**Table 5 nutrients-14-02108-t005:** Relationship between genotypes of *MTHFR*, *MTRR*, *MTR*, *TYMS,* and SGA.

Genotype		Controls [*n*(%)]	SGA [*n*(%)]	OR (95% CI)	*p*
*MTHFR* C677T	CC	80(93.02)	6(6.98)	1	-
	CT + TT	580(94.93)	31(5.07)	0.71(0.29–1.76)	0.463
*MTHFR* A1298C	AA	521(94.56)	30(5.44)	1	-
	AC + CC	139(95.21)	7(4.79)	0.88(0.38–2.03)	0.756
*MTRR* A66G	AA	381(93.84)	25(6.16)	1	-
	AG + GG	279(95.88)	12(4.12)	0.66(0.32–1.33)	0.241
*MTR* A2756G	AA	534(94.85)	29(5.15)	1	-
	AG + GG	126(94.03)	8(5.97)	1.17(0.52–2.62)	0.704
*TYMS* rs3819102	AA	411(93.62)	28(6.38)	1	-
	AG + GG	249(96.51)	9(3.49)	0.53(0.25–1.14)	0.106

*MTHFR*, methylenetetrahydrofolate reductase; *MTRR*, methionine synthase reductase; *MTR*, methionine synthase; *TYMS*, thymidylate synthetase; SGA, small-for-gestational-age.

## Data Availability

The datasets generated or analyzed during the current study are not publicly available due to the data management requirements of our institution, but are available from the corresponding author on reasonable request.

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
