# Peer review of "Relationships between Maternal Gene Polymorphisms in One Carbon Metabolism and Adverse Pregnancy Outcomes: A Prospective Mother and Child Cohort Study in China"

_nutrients, 2022, doi:10.3390/nu14102108_

Round 1

Reviewer 1 Report

Although the study is rigorous and bring some interesting new features, I think that the authors have chosen a non appropriate journal. No food, diet or other similar are contained in manuscript. No referring to diet it is contained in tables, in phenotype assesment, so i suppose, nether in the questionnaires given to the women. I think this manuscript is better for a molecular genetics journal instead of one of nutrition.

In case the editor would like to approve this manuscript I have some improvement proposal:

  1. In Introduction must to be clear the motivation of polymorphism choosing. At moment, in introduction, I see only the motivation linked to reference #14 but is poor. In discussion it is present a clear motivation: i think that is better to move part of it in introduction.
  2. Today it is widely known that a gene can be modified in its expression by both genetics and epigenetics factors and when bot are take in consideration, often the polymorphisms are more important than epigenetic (Naselli et al.: "Role and importance of polymorphisms with respect to DNA methylation......". Gene. 2014 Feb 15;536:29-39) So, in 2022 it is no admissible that a manuscript don't give data of epigenetic epigenomic status together to the genetic results. In absence, it is important to give motivations, as weakness of research, also by citing the above reported reference to attempt to give more importance to the existing data.
  3. All the gene acronyms must to be reported in italics and uppercase as required by international guidelines.

Author Response

Point 1: In Introduction must to be clear the motivation of polymorphism choosing. At moment, in introduction, I see only the motivation linked to reference #14 but is poor. In discussion it is present a clear motivation: i think that is better to move part of it in introduction. Response: Revised, please see lines 59-65 . High-level folate was associated with lower risks of the above adverse pregnancy outcomes [14,15]. Single nucleotide polymorphisms (SNPs) related folic acid metabolic enzymes in birth defects had been identified [16]. Recently, both Indian and Japanese studies suggested that folic acid metabolic enzymes related SNPs may be related to the risks of PTB and LBW [17,18]. Point 2: Today it is widely known that a gene can be modified in its expression by both genetics and epigenetics factors and when bot are take in consideration, often the polymorphisms are more important than epigenetic (Naselli et al.: "Role and importance of polymorphisms with respect to DNA methylation......". Gene. 2014 Feb 15;536:29-39) So, in 2022 it is no admissible that a manuscript don't give data of epigenetic epigenomic status together to the genetic results. In absence, it is important to give motivations, as weakness of research, also by citing the above reported reference to attempt to give more importance to the existing data. Response: Revised (lines 260-264). Besides, this study didn’t analyze epigenetics of study population, which could modify the gene expression [48] and the interaction between gene polymorphism and epigenetics may alter the activity of folic acid-related enzymes. However, adequate folic acid status of those pregnant women may maintain similar methylation level of those subjects, which may reduce the impacts of epigenetics. Point 3: All the gene acronyms must to be reported in italics and uppercase as required by international guidelines Response: Revised.

Reviewer 2 Report

The study subject is welcomed due to its potentially future aid regarding prevention of prematurity and low birth weight and therefore neonatal specific morbidity.

However, your study needs several clarifications. 

regarding study group exclusions - what about other pregnancy associated pathology such as diabetes, hypertension, thrombophilia, high BMI - shouldn't also these be excluded as they can bias the results?

even in introduction you mentioned environmental factors proved to be correlated with PTB - maternal education level, gestational age, alcohol and 54 smoking consumption. Are these patients excluded from your analysis? - if yes, this need to be clearly specified.

you need to describe more specifically how subgroups of cases and controls were created.

line 199 - H Guo raised hypothesis that folic acis suplimentation has protective effect regarding SGA after 37 weeks after exclusions of multiparous which were considered to offer a more favourable environment for placental and fetal growth. The 37 weeks subgroups need to be clearly specified.

you mentioned that you included women up to 45 years old. What about the known risk of PTB related to age? This possible bias needs to be clearly specified. please see below

Fuchs F, Monet B, Ducruet T, Chaillet N, Audibert F. Effect of maternal age on the risk of preterm birth: A large cohort study. PLoS One. 2018 Jan 31;13(1):e0191002. doi: 10.1371/journal.pone.0191002.

Author Response

Point 1: regarding study group exclusions - what about other pregnancy associated pathology such as diabetes, hypertension, thrombophilia, high BMI - shouldn't also these be excluded as they can bias the results?

Response: Women with diabetes, hypertension and thrombophilia were excluded at recruitment, and has been specified in the manuscript (lines 83).The range of pre-pregnancy BMI was 16.0-39.3 kg/m2. We included pre-pregnancy BMI as a potential confounding factor in the model to control its potential influence and it turned out to be no associations.

Point 2: even in introduction you mentioned environmental factors proved to be correlated with PTB - maternal education level, gestational age, alcohol and 54 smoking consumption. Are these patients excluded from your analysis? - if yes, this need to be clearly specified.

Response: We included maternal education level and maternal age at childbearing as potential confounding factors in the model to control their potential effects. The proportions of pregnant women who smoked and drank were 0.6% and 1.3% respectively, and the low prevalence of smoking and drinking may have little impacts on the overall associations. No association was found in the subgroup analyses excluding women who smoking and drinking currently.

Point 3: you need to describe more specifically how subgroups of cases and controls were created.

Response: Revised ( lines 130-135).

Point 4: line 199 - H Guo raised hypothesis that folic acis suplimentation has protective effect regarding SGA after 37 weeks after exclusions of multiparous which were considered to offer a more favourable environment for placental and fetal growth. The 37 weeks subgroups need to be clearly specified.

Response 4: The proportion of SGA after 37 gestational weeks was 91.9% (34/37). In the sensitive analyses of “37 weeks subgroups”, there were no sighnificant associations between MTHFR C677T, MTHFR A1298C, MTRR A66G, MTR A2756G, TYMS rs3819102 and PTD, LBW, SGA, which was consistent with the full analysis.

Point 5: you mentioned that you included women up to 45 years old. What about the known risk of PTB related to age? This possible bias needs to be clearly specified. please see below

Fuchs F, Monet B, Ducruet T, Chaillet N, Audibert F. Effect of maternal age on the risk of preterm birth: A large cohort study. PLoS One. 2018 Jan 31;13(1):e0191002. doi: 10.1371/journal.pone.0191002.

Response 5: The range of maternal age was 16.0-44.1 years old, and we included maternal age at childbearing as a confounding factor in the model to control its potential relationship with adverse birth outcomes.

Round 2

Reviewer 1 Report

Although the authors have modified the manuscript soundly with the reviewer suggestions, the entire work results slightly sufficient good to publication in this "food" journal.

Reviewer 2 Report

Questions and suggestions were explained and therefore study can be taken into account for publication